

# Technical Note: Preparation and purification of atmospherically relevant α-hydroxynitrate esters of monoterpenes

Elena Ali McKnight, Nicole P. Kretekos, Demi Owusu, Rebecca Lyn LaLonde

Chemistry Department, Reed College, Portland, OR, 97202, USA

*Correspondence to*: Rebecca Lyn LaLonde (rlalonde@reed.edu)

**Abstract.** Organic nitrate esters are key products of terpene oxidation in the atmosphere. We report here the preparation and purification of nine nitrate esters derived from (+)-(3)-carene, limonene, α-pinene, β-pinene and perillic alcohol. The availability of these compounds will enable detailed investigations into the structure

reactivity relationships of aerosol formation and processing and will allow individual investigations into aqueous phase reactions of organic nitrate esters.

## 1 Introduction

Biogenic volatile organic compound (BVOC) emissions account for ~88% of non-methane VOC emissions. Of the total BVOC estimated by the Model of Emission of Gases and Aerosols from Nature version 2.1

(MEGAN2.1), isoprene is estimated to comprise half, and methanol, ethanol, acetaldehyde, acetone, α-pinene, β-pinene, limonene, ethene and propene together encompass another 30%. Of the terpenoids, α-pinene alone is estimated to generate ~66 Tg year$^{-1}$ (Guenther et al., 2012). These monoterpenes can be oxidized by nitrate radicals that are projected to account for more than half of the monoterpene-derived secondary organic aerosol (SOA) in the US (Pye et al., 2010). This nitrate oxidation pathway has been shown to be important with 30-40%

of monoterpene emissions occurring at night (Pye et al., 2010).

The full role of organic nitrates (ON) is complicated with many different sources and sinks (Perring et al., 2013). Fully deconvoluting the atmospheric processing of terpene-derived organic nitrates is difficult particularly due to partitioning into the aerosol phase in which hydrolysis and other reactivity can occur (Bleier

and Elrod, 2013; Rindelaub et al., 2014; 2015; Romonosky et al., 2015; Thomas et al., 2016). Hydrolysis reactions of nitrate esters of isoprene have been studied directly (Jacobs et al., 2014) and the hydrolysis of

organonitrates have been studied in bulk (Baker and Easty, 1950). These and other studies have shown that the hydrolysis of organonitrates is dependent on structure (Darer et al., 2011). For example, primary and secondary organic nitrates are thought to be relatively stable (Hu et al., 2011). In contrast, tertiary nitrates have been shown to hydrolyze on the order of hours (Boyd et al., 2015; Liu et al., 2012) to minutes (Darer et al., 2011). To the

best of our knowledge there is only one study of the hydrolysis of an isolated terpene-derived hydroxynitrate (**2**) (Rindelaub et al., 2016).



**Figure 1: Two hydroxynitrate esters with available spectral data. Relative stereochemistry is undefined.**

Furthermore, fully understanding the atmospheric processes of organic molecules is restricted by the ability to

identify these species (Nozière et al., n.d.). Part of this challenge is, of course, related to the lack of available standards.  While one certainly cannot synthesize all of the atmospherically relevant organic nitrates, having access to representative compounds from monoterpenes would enable key studies.  With these molecules in hand, the atmospheric chemistry community could directly study the organic nitrate reactivity, such as hydrolysis, and deconvolute the structure-reactivity relationships. Additionally, novel method development

would be enabled and validated (Rindelaub et al., 2016b).  For example, Nozière and co-authors called attention to "the lack of NMR spectra libraries for atmospheric markers" as a barrier for utilizing NMR spectroscopy atmospheric science. While the preparation of α-hydroxynitrates of terpenes has been alluded to in a handful of reports, we have only been able to identify NMR data for two species (Figure 1). Two main methods appear to have been attempted, treating an epoxide with either fuming nitric acid (Rollins et al., 2010) or bismuth nitrate

(Rindelaub et al., 2016b; Romonosky et al., 2015). The utility of the former is limited by the extreme hazards involved with mixing fuming nitric acid with organic materials (Univ. of Calif., Berkeley, 2018) and only provides characterization data for a single compound. On the surface the latter appears to be a usable method but on closer inspection lacks spectral data, perhaps due to intractable and inseparable mixtures (Romonosky et al., 2015) and has not been without controversy (Ma et al., 2011; Pöschl et al., 2011).


## 2 Experimental

### 2.1 Instrumentation and materials

All starting materials and reagents were purchased from commercial sources and used without further purification unless otherwise noted. Bismuth nitrate (98%) was purchased from Strem Chemicals and ground to a fine powder prior to use. Isoprene, MVK, (+)-(3)-carene, β-pinene, α-pinene oxide and 1,2-limonene oxide were purchased from Aldrich. Silica gel chromatography was performed on a TeldyneIsco CombiFlash+ Lumen using 25 μm SiliCycle spherical silica gel. $^1$H and $^{13}$C NMR spectra were recorded on a Bruker AV400 spectrometer with chloroform (7.26 ppm) or benzene (7.28 ppm) as internal standards. IR spectra were recorded on a ThermoNicolet IR100 spectrometer using a Thunderdome Attenuated Total Reflectance (ATR) sample accessory. Melting points were collected on a DigiMelt 160 melting point apparatus. Exact masses were collected at the BioAnalytical Mass Spectrometry facility at Portland State University using a ThermoElectron LTQ-Orbitrap Discovery high resolution mass spectrometer with a dedicated Accela HPLC system. Abbreviations include hexanes (Hex), ethyl acetate (EtOAc), singlet (s), doublet (d), doublet of doublets (dd), triplet (t), multiplet (m), broad (b), weak (w), medium (m), strong (s).

### 2.2 Preparation of epoxides

#### 2.2.1 *trans*-Carene oxide (3)

To a solution of carene (0.86 mL, 1 g, 7.34 mmol) in dichloromethane (25 mL) at 0 ºC was added m-CPBA (1.85 g, 8.1 mmol, 1.1 equiv). The solution was warmed to 23 ºC over 1 hour. The solution was poured onto sat. aq sodium bicarbonate (30 mL) and extracted with dichloromethane (3 x 30 mL). The combined organics were washed with sat. aq sodium bicarbonate (2 x 30 mL), dried (MgSO$_4$) and concentrated to yield a crude clear, colorless oil (1.17 g). The crude oil was purified by column chromatography (SiO$_2$; 0 – 25% EA/Hex over 10 column volumes) to yield clear colorless oil (685 mg, 4.5 mmol; 61% yield): $^1$H NMR (400 MHz, Chloroform-d) δ 2.82 (s, 1H), 2.28 (ddd, J = 16.4, 9.0, 1.9 Hz, 1H), 2.13 (dd, J = 16.2, 9.1 Hz, 1H), 1.63 (dt, J = 16.4, 2.3 Hz, 1H), 1.48 (dd, J = 16.1, 2.3 Hz, 1H), 1.24 (s, 3H), 1.00 (s, 3H), 0.72 (s, 3H), 0.52 (td, J = 9.1, 2.3 Hz, 1H), 0.44 (td, J = 9.1, 2.3 Hz, 1H) ppm. Spectral data is consistent with literature reports (Cabaj et al., 2019).

#### 2.2.2 *cis*-Carene oxide (6)

According to the methods of Crocker, a round bottom flask was charged with a solution of (+)-(3)-carene (2.7 g, 20 mmol) in dioxane (20 mL) and water (10 mL). Calcium carbonate (2 g, 20 mmol) and N-bromosuccinimide



(7 g, 40 mmol) were added to the solution. The internal temperature rose to 50 ºC after the initial addition. The mixture was stirred for 2 h then poured onto water (50 mL), filtered, and washed with diethyl ether. The filtrate was extracted with ether (2 x 100 mL). The combined extract was washed with water (3 x 100 mL), sodium thiosulfate (5% aq, 20 mL), dried (Na$_2$SO$_4$) and concentrated to yield a crude pale oil. Purification by column

chromatography (SiO$_2$; 0 – 50% EA/Hex) yielded the bromohydrin as a white crystalline solid (4.6 g, 99% yield): [1]H NMR (400 MHz, Chloroform-d) δ 4.07 (dd, J = 11.1, 7.6 Hz, 1H), 2.50 – 2.36 (m, 2H), 2.21 (dd, J = 14.6, 10.1 Hz, 1H), 1.41 (dd, J = 4.9, 1.2 Hz, 1H), 1.38 (s, 3H), 1.03 (s, 3H), 1.00 (s, 3H), 0.91 – 0.80 (m, 1H), 0.70 (m, 1H). Spectral data is consistent with literature reports (Cocker, 1969). Bromohydrin (2.44 g, 10.5 mmol) was dissolved in 100 mL tBuOH (warmed with a water bath). Potassium *tert*-butoxide (2.44 g, 21.8

mmol, 2.08 equiv) was added and the solution was stirred at rt for 2 hours (9 am – 11 am). The solution was poured onto water (50 mL), extracted with diethyl ether (3 x 75 mL), washed with water (3 x 50 mL), dried (MgSO$_4$) and concentrated to yield 2.5 g crude clear oil. Purification by column chromatography (40 g SiO$_2$; 0 – 25% EA/Hex over 10 CVs) yielded a clear colorless oil (1.31 g, 63% yield): [1]H NMR (400 MHz, Chloroform-d) δ 2.89 (d, J = 5.6 Hz, 1H), 2.30 (ddd, J = 16.7, 9.0, 5.6 Hz, 1H), 2.08 (dd, J = 16.4, 8.9 Hz, 1H), 1.82 (s, 1H),

1.78 (s, 1H), 1.32 (s, 3H), 0.98 (s, 3H), 0.94 (s, 3H), 0.63 – 0.49 (m, 2H). Spectral data is consistent with literature reports (Cocker and Grayson, 1969).

### 2.2.3 8,9-Limonene oxide (9)

Methyl vinyl ketone (10 mL, 90 mmol), isoprene (10 mL, 75 mmol), and DCM (90 mL) were added to a round bottom flask with stir bar. The flask was purged with inert atmosphere and chilled to 0°C with stirring. AlCl$_3$

(1.2 g, 9 mmol) was added in three portions over 10 minutes. The ice bath was removed and the reaction mixture was stirred for 1.5 hours. The crude reaction mixture was filtered through a 1.5 inch silica gel pad (350 mL, 9 cm diameter) washed with 8% Ethyl Acetate:Hexanes (4 x 160 mL). The filtrate was concentrated to yield 1-(4-methyl-3-cyclohexene) ethenone (**8**) as a clear yellow liquid (9.68 g, 88% yield): [1]H NMR (400 MHz, Chloroform-d) δ 5.44 – 5.38 (m, 1H), 2.59 – 2.49 (m, 1H), 2.15 – 2.20 (s, 4H), 2.06 – 1.95 (m, 3H), 1.67 (s,

3H), 1.61 (m, 2H). Spectral data is consistent with literature reports (Buss et al., 1987). Sodium hydride (1.766 g, 44.16 mmol) was suspended in dry DMSO (25 mL). A solution of trimethylsulfonium iodide (10.356 g, 47.06 mmol) in DMSO (50 mL) was added via cannula. The solution was stirred at rt for 30 minutes then heated to 50 ºC until gas evolution ceased (1 h). To this solution was added 1-(4-methyl-3-cyclohexen-1-yl) ethenone **8** (5 g, 36 mmol) and heated to 70 ºC for 2 hours. The reaction mixture was cooled to room temperature and poured

onto sat. aq. NH$_4$Cl (100 mL), extracted with MTBE (4 x 100 mL). The combined organics were washed with



water (4 x 50 mL), dried (MgSO₄) and concentrated to yield 5.1 g crude oil. Purification by column chromatography (80 g SiO₂; 0-20% EA/Hex over 10 CV) yielded 8,9-limonene oxide as a clear, colorless oil (3.76 g, 68% yield): $^1$H NMR (400 MHz, Chloroform-d) δ 5.43 – 5.33 (br s, 1H), 2.67 (t, J = 5.3 Hz, 1H), 2.58 (dd, J = 10.9, 4.8 Hz, 1H), 2.17 – 1.73 (m, 5H), 1.66 (s, 3H), 1.44 (m, 2H), 1.29 (d, J = 4.5 Hz, 3H). Spectral data is consistent with literature reports (Almeida and Jr., 2005)

### 2.2.4 *cis*-1,2-Limonene oxide (12-*cis*)

According to the methods of Steiner, a 20 mL scintillation vial was charged with 1,2-limonene oxide, pyrrolidine & water. The vial was sealed with a Teflon lined cap and heated to 88 ºC for 24 h. The vial was cooled to rt, transferred to a separatory funnel with 100 mL pentane, washed with sat. aq. ammonium chloride (3 x 30 mL) and water (1 x 50 mL). The organics were dried (MgSO₄) and concentrated to yield a light yellow oil. Purification by column chromatography (80 g spherical SiO₂; 0-15% EA/Hex over 10 CV's) yielded *cis*-1,2-limonene oxide as a clear, colorless oil. (1.52 g, 64%): $^1$H NMR (400 MHz, Chloroform-d) δ 4.77 – 4.71 (m, 1H), 4.69 (s, 1H), 3.07 (s, 1H), 2.22 – 2.07 (m, 2H), 1.90 – 1.84 (m, 2H), 1.75 – 1.64 (m, 5H), 1.33 (s, 3H), 1.28 – 1.15 (m, 1H) ppm. Spectral data is consistent with literature reports (Steiner et al., 2002).

### 2.2.5 *trans*-1,2-Limonene oxide (12-*trans*)

According to the methods of Steiner, a 50 mL flask was charged with 1,2-limonene oxide (4.57 g, 30 mmol), pyrazole (0.34 g, 5 mmol) and water (16.2 mL). The reaction was heated to reflux for 5 hours. The reaction mixture was cooled to 80 ºC and transferred to a separatory funnel. Warm water (80 ºC; 60 mL) was added. The emulsion was extracted with pentane (3 x 50 mL). The combined organic layers were dried (MgSO₄) and concentrated to yield 1.481 g crude oil. Purification by column chromatography (80 g spherical SiO₂; 0 – 20% EtOAc/Hex over 13.5 CV) yielded *trans*-1,2-limonene oxide (485 mg, 20% yield): $^1$H NMR (400 MHz, Chloroform-d) δ 4.68 (s, 2H), 3.01 (d, J = 5.3 Hz, 1H), 2.11 – 2.00 (m, 2H), 1.96 – 1.82 (m, 1H), 1.75 – 1.64 (m, 5H), 1.45 – 1.36 (m, 2H), 1.34 (s, 3H) ppm. Spectral data is consistent with literature reports (Steiner et al., 2002).

### 2.2.6 Perillic alcohol epoxide (15)

Perillic alcohol (5 mL, 31.46 mmol) was combined with dichloromethane (62 mL) in a round bottom flask with stir bar and cooled to 0 ºC under inert atmosphere. m-CPBA (5.97 g, 34.61 mmol) was added in 5 portions over the course of 10 minutes. After 30 minutes the reaction was warmed back up to room temperature and allowed



to stir for an hour at room temperature. The reaction mixture was filtered through celite, washed with DCM (3 x 30 mL), then transferred to a separatory funnel where it was washed with sodium bicarbonate (50 mL) and brine (50 mL). The organic layer was dried (Na₂SO₄) and concentrated to yield a crude oil (5.1 g) Purification by column chromatography (80 g spherical SiO₂; 0 – 100% EtOAc/Hex over 15 CV's) yielded the title compound

as a clear, colorless oil (3.363 g, 19.8 % yield): ¹H NMR (400 MHz, Chloroform-*d*) δ 4.77 – 4.66 (m, 2H), 3.78 – 3.59 (m, 2H), 3.41 – 3.31 (m, 1H), 2.26 – 2.14 (m, 1H), 2.09 (dtd, *J* = 14.9, 5.6, 1.8 Hz, 1H), 1.88 – 1.83 (m, 1H), 1.80 – 1.59 (m, 7H), 1.56 – 1.38 (m, 1H), 1.32 – 1.16 (m, 1H). Spectral data is consistent with literature reports (Thomas et al., 2016).

**2.2.7 β -pinene oxide**

Potassium peroxide monosulfate (4.750 g, 31.20 mmol) was dissolved in deionized water (60 mL). Sodium bicarbonate (4.019 g, 47.84 mmol) was placed into a 125 mL erlenmeyer flask and acetone (40 mL) was added, followed by β-pinene (1.59 mL, 9.974 mmol). The solution of potassium peroxide monosulfate mixture was slowly added into the β-pinene mixture over the course of three minutes via syringe while the mixture was

stirring at 800 rpm. The reaction mixture was stirred for exactly 30 minutes while stirring at 800 rpm at room temperature. The reaction mixture was transferred to a separatory funnel and extracted with dichloromethane (2 x 40 mL), dried (MgSO₄) and concentrated to produce a clear oil (1.500 g, 96% yield). ¹H NMR (400 MHz, Chloroform-*d*) δ 2.79 (d, *J* = 4.8 Hz, 1H), 2.62 (d, *J* = 4.8 Hz, 1H), 2.30 (dtd, *J* = 10.3, 5.9, 1.5 Hz, 1H), 2.19 (ddd, *J* = 15.1, 10.7, 8.1 Hz, 1H), 2.06 – 1.97 (m, 1H), 1.97 – 1.79 (m, 2H), 1.75 – 1.65 (m, 2H), 1.53 (t, *J* = 5.4

Hz, 1H), 1.27 (s, 3H), 0.94 (s, 3H). Spectral data is consistent with literature reports (Charbonmeau et al, 2018).

**2.3 General nitration method**

A round bottom flask was charged with a solution of epoxide (2 mmol) in toluene, dioxane or dichloromethane (10 mL, 0.2 M). Bismuth nitrate (1.164 g, 2.4 mmol, 1.2 equiv) was added to the reaction mixture.  The reaction mixture was stired for 30 – 60 minutes. When TLC indicated complete consumption of starting material the

reaction was filtered through a 1 inch celite pad and washed with DCM (2 x 15 mL). The filtrate was transferred to a separatory funnel and washed with sodium bicarbonate (3 x 15 mL). The organic layer was dried with sodium sulfate, filtered and concentrated to yield a crude liquid. Desired products were isolated via column chromatography.



### 2.3.1 Nitrate ester 4

Nitration of *trans*-(3)-carene (3 mmol) was carried out according to the general method with dioxane. Purification of the crude oil by column chromatography(12 g SiO$_2$, 0-30% EtOAc/Hex) yielded the title compound as a white crystalline solid (345 mg, 53% yield): mp = 52.2-54.0 °C; IR (ATR) cm$^{-1}$ 3412 (bd w, OH), 2944 (w), 2360 (w), 1616 (vs, NO$_2$), 1291 (m, NO$_2$), 868 (m, NO$_2$); $^1$H NMR (400 MHz, Chloroform-*d*) δ 3.70 (dd, *J* = 9.6, 7.6 Hz, 1H), 2.86 – 2.78 (m, 1H), 2.16 (dd, *J* = 14.8, 7.5 Hz, 1H), 1.83 (ddd, *J* = 14.8, 9.6, 7.9 Hz, 1H), 1.65 (s, 3H), 1.47 (dd, *J* = 14.3, 4.1 Hz, 1H), 1.03 (s, 3H), 1.01 (s, 3H), 0.83 – 0.73 (m, 2H); $^{13}$C NMR (101 MHz, Chloroform-*d*) δ 95.68, 71.03, 28.37, 28.34, 27.02, 19.95, 18.98, 17.95, 15.70, 14.39 ppm.

### 2.3.2 *cis*-4-caranone (7)

Nitration of *cis*-(3)-carene was attempted according to the general method with dioxane or dichloromethane. Purification of the crude oil by column chromatography (12 g SiO$_2$, 0-40% EtOAc/Hex) yielded *cis*-4-caranone as a clear, colorless oil (53% yield) R$_f$ 0.51 (5% EtOAc:Hex; anisaldehyde); $^1$H NMR (400 MHz, Chloroform-*d*) δ 2.54 (ddd, *J* = 18.0, 8.4, 1.0 Hz, 1H), 2.44 – 2.25 (m, 3H), 1.33 – 1.20 (m, 1H), 1.14 – 0.99 (m, 5H), 0.97 (d, *J* = 6.4 Hz, 3H), 0.86 (s, 3H); $^{13}$C NMR (101 MHz, Chloroform-*d*) δ 216.79, 41.99, 36.84, 29.77, 27.91, 22.83, 20.34, 19.47, 14.89, 14.11 ppm. Spectral data is consistent with literature reports (Kolehmainen et al., 1993).

### 2.3.3 Reaction of 8,9-limonene oxide (9).

Nitration of 8,9-limonene oxide (**9**) was attempted according to the general method with acetonitrile. Acetonitrile yielded limonene-8,9-diol as a crude oil (87% yield): R$_f$ 0.81 (15% EtOAc:Hex; anisaldehyde); $^1$H NMR (400 MHz, Chloroform-*d*) δ 5.39 – 5.30 (m, 1H), 4.11 – 4.01 (m, 1H), 3.78 (ddt, *J* = 13.9, 8.4, 1.7 Hz, 1H), 2.14 – 1.85 (m, 5H), 1.72 – 1.58 (m, 5H), 1.27 – 1.13 (m, 4H), 1.12 – 1.01 (m, 1H) ppm. Purification by column chromatography (12 g SiO$_2$ gold column, 0-50% EtOAc/Hex over 30 CV) yielded 4-Dimethyl-3-cyclohexene-1-acetaldehyde as a clear, colorless oil (105 mg, 35% yield): $^1$H NMR (400 MHz, Chloroform-d) δ 9.69 (d, J = 5.5 Hz, 1H), 5.42 – 5.35 (m, 1H), 2.31 (qd, J = 7.0, 6.0, 2.4 Hz, 1H), 2.03 (s, 4H), 1.68 – 1.64 (m, 5H), 1.39 (dddd, J = 12.9, 10.6, 9.2, 5.3 Hz, 1H), 1.09 (d, J = 7.0, 5.2 Hz, 3H) ppm. Spectral data is consistent with literature reports (Uehara et al., 2017).





### 2.3.4 Nitrate ester 13

Nitration of *cis*-1,2-limonene oxide was carried out according to the general method with dioxane or dichloromethane. Purification by column chromatography (12 g SiO₂ gold column, 0-50% EtOAc/Hex over 30 CV) yielded the title compound as a clear, colorless oil (53% in dioxane; 62% in DCM): IR (ATR) cm$^{-1}$ 3660 (w, alcoholic OH), 3340 (bd w, alcoholic OH), 2980 (m), 1618 (s, NO₂), 1291 (m, NO₂), 860 (vs, NO₂); $^1$H NMR (400 MHz, Chloroform-d) δ 4.76 (dd, J = 8.1, 1.5 Hz, 2H), 4.13 (s, 1H), 2.35 (ddt, J = 11.6, 8.4, 4.1 Hz, 1H), 2.24 (dtd, J = 14.8, 3.7, 1.4 Hz, 1H), 1.98 – 1.78 (m, 3H), 1.76 – 1.73 (m, 4H), 1.68 – 1.58 (m, 4H), 1.49 (tdd, J = 13.3, 11.6, 3.6 Hz, 1H); $^{13}$C NMR (101 MHz, Chloroform-*d*) δ 148.49, 109.48, 91.34, 69.22, 36.84, 33.86, 29.90, 25.77, 20.98, 20.93 ppm.

### 2.3.5 Nitrate ester 14

Nitration of *trans*-1,2-limonene oxide was carried out according to the general method with dioxane or dichloromethane. Purification by column chromatography (12 g SiO₂, 0-50% EtOAc/Hex over 30 CV) yielded the title compound as a clear, colorless oil (63% in dioxane; 54% in DCM): Rf 0.39 (15% EtOAc:Hex; anisaldehyde); IR (ATR) cm$^{-1}$ 3414 (bd w, alcoholic OH), 2940 (w), 1627 (s, NO₂), 1279 (m, NO₂), 876(m, NO₂), 851 (m, NO₂); $^1$H NMR (400 MHz, Chloroform-d) δ 5.01 (s, 1H), 4.76 (d, J = 6.6 Hz, 2H), 2.19 (tt, J = 10.9, 3.1 Hz, 1H), 2.03 (ddd, J = 14.7, 12.1, 2.6 Hz, 1H), 1.92 (dq, J = 14.4, 3.5, 2.7 Hz, 1H), 1.74 (s, 3H), 1.69 – 1.56 (m, 4H), 1.32 (s, 3H); $^{13}$C NMR (101 MHz, Chloroform-d) δ 148.27, 109.57, 84.43, 69.55, 37.96, 34.78, 29.91, 26.96, 20.89 ppm.

### 2.3.6 Nitration of perillic alcohol epoxide (15)

Nitration of 4-(1-methylethenyl)-7-Oxabicyclo[4.1.0]heptane-1-methanol (**15**) was carried out according to the general method with dichloromethane. Purification by column chromatography (12 g SiO₂, 0-40% EtOAc/Hex over 30 CV) yielded the following two regioisomers, nitrate esters **16** and **17**.

### 2.3.6.1 Nitrate ester 16

The title compound was isolated as a white, crystalline solid (106 mg, 23% yield): mp = 72-75°C; Rf 0.31 (30% EtOAc:Hex; anisaldehyde); IR (ATR) cm$^{-1}$ 3285 (bd w, alcoholic OH), 2941 (w), 1627 (vs, NO₂), 1280 (s, NO₂), 880 (s, NO₂), 850 (s, NO₂); $^1$H NMR (400 MHz, Chloroform-*d*) δ 5.23 (s, 1H), 4.80 – 4.73 (m, 2H), 3.74 (dd, *J* = 10.9, 5.0 Hz, 1H), 3.44 (dd, *J* = 11.0, 5.0 Hz, 1H), 2.59 (d, *J* = 1.0 Hz, 1H), 2.27 – 2.16 (m, 1H), 2.03 –


1.97 (m, 2H), 1.79 (t, $J$ = 5.2 Hz, 1H), 1.71 – 1.61 (m, 3H), 1.58 (d, $J$ = 0.8 Hz, 4H); $^{13}$C NMR (101 MHz, Chloroform-$d$) δ 148.24, 109.58, 79.89, 70.98, 66.45, 38.47, 29.70, 29.55, 25.10, 20.77 ppm.

### 2.3.6.2 Nitrate ester 17

The title compound was isolated as a clear, colorless oil (149 mg, 32% yield): $R_f$ 0.20 (30% EtOAc:Hex; anisaldehyde); IR (ATR) cm$^{-1}$ 3286 (bd w, alcoholic OH), 2941 (w), 1623 (vs, NO$_2$), 1288 (s, NO$_2$), 863 (vs, NO$_2$); HRMS: calc'd for $C_{10}H_{16}O_4N^-$ (214.10813, observed; 214.10738, expected); $^1$H NMR (400 MHz, Chloroform-$d$) δ 4.76 (d, $J$ = 8.5 Hz, 2H), 4.31 (s, 1H), 4.27 (d, $J$ = 13.0 Hz, 1H), 3.99 (d, $J$ = 13.0 Hz, 1H), 2.41 (tt, $J$ = 12.1, 3.6 Hz, 1H), 2.27 – 2.19 (m, 1H), 1.98 – 1.78 (m, 3H), 1.75 (s, 3H), 1.72 – 1.65 (m, 1H), 1.51 – 1.38 (m, 1H); $^{13}$C NMR (101 MHz, Chloroform-$d$) δ 148.58, 109.43, 92.51, 65.67, 62.91, 37.47, 33.53, 25.33, 25.26, 20.86 ppm.

### 2.3.7 Nitration of α-pinene

Nitration of alpha-pinene oxide was carried out according to the general method. Crude oil from a 10 mmol reaction was purified by column chromatography (40 g SiO$_2$, 0-40% EtOAc/Hex over 30 CV) yielded the following isomerization products and nitrate esters.

### 2.3.7.1 α-campholenic aldehyde (19)

The title compound was isolated as a clear, colorless oil: $R_f$ 0.59 (10% EtOAc:Hex; anisaldehyde); $^1$H NMR (400 MHz, Benzene-$d_6$) δ 9.51 (t, $J$ = 2.1 Hz, 1H), 5.23 (s, 1H), 2.41 (dddt, $J$ = 13.8, 6.0, 2.7, 1.6 Hz, 1H), 2.25 – 2.16 (m, 1H), 2.11 (ddd, $J$ = 15.9, 4.4, 1.8 Hz, 1H), 2.00 (ddd, $J$ = 15.9, 10.3, 2.3 Hz, 1H), 1.80 (ddp, $J$ = 15.8, 9.2, 2.5 Hz, 1H), 1.58 (dt, $J$ = 3.0, 1.6 Hz, 3H), 0.90 (s, 3H), 0.68 (s, 3H). Spectral data is consistent with literature reports (Thomas et al., 2016).

### 2.3.7.2 (1S)-2-Methyl-5-(1-methylethylidene)-2-cyclohexen-1-ol (20)

The title compound was isolated as a white, crystalline solid: $R_f$ 0.47 (20% EtOAc:Hex; anisaldehyde); $^1$H NMR (400 MHz, Chloroform-$d$) δ 5.49 (ddd, $J$ = 4.4, 3.2, 1.5 Hz, 1H), 3.99 (t, $J$ = 4.2 Hz, 1H), 2.86 (d, $J$ = 19.6 Hz, 1H), 2.69 – 2.56 (m, 2H), 2.35 (d, $J$ = 15.5 Hz, 1H), 1.78 – 1.75 (m, 3H), 1.74 (s, 3H), 1.68 (s, 3H); $^{13}$C NMR (101 MHz, Chloroform-$d$) δ 135.94, 125.57, 124.23, 123.29, 70.34, 35.92, 29.83, 20.31, 20.14, 19.91 ppm. Spectral data is consistent with literature reports (Motherwell et al., 2004).





### 2.3.7.3 *trans*-Carveol (21)

The title compound was isolated as a clear, colorless oil: $R_f$ 0.37 (20% EtOAc:Hex; anisaldehyde); [1]H NMR (400 MHz, Chloroform-*d*) δ 5.62 (dd, $J$ = 5.2, 1.5 Hz, 1H), 4.78 – 4.73 (m, 2H), 4.06 – 4.03 (m, 1H), 2.34 (ddd, $J$ = 13.7, 9.8, 2.9 Hz, 1H), 2.21 – 2.12 (m, 1H), 1.96 (dq, $J$ = 13.6, 2.3 Hz, 1H), 1.93 – 1.85 (m, 1H), 1.85 – 1.81 (m, 3H), 1.77 (s, 3H), 1.64 (dd, $J$ = 13.2, 4.0 Hz, 1H), 1.61 – 1.57 (m, 1H) ppm. Spectral data is consistent with literature reports). (Motherwell et al., 2004).

### 2.3.7.4 *trans*-Carveol nitrate ester 22

The title compound was isolated as a clear, colorless oil: : IR (ATR) cm[-1] 3353 (bd w, alcoholic OH), 2917 (w), 1614 (vs, $NO_2$), 1293 (s, $NO_2$), 868 (m, $NO_2$); [1]H NMR (400 MHz, Chloroform-*d*) δ 5.32 (ddd, $J$ = 9.9, 3.5, 2.1 Hz, 1H), 4.12 (ddd, $J$ = 10.6, 4.6, 1.9 Hz, 1H), 2.68 – 2.49 (m, 2H), 1.88 (t, $J$ = 4.8 Hz, 1H), 1.43 (dd, $J$ = 14.1, 3.7 Hz, 1H), 1.24 (dd, $J$ = 13.8, 4.8 Hz, 1H), 1.11 (s, 3H), 0.97 (s, 3H), 0.94 (s, 3H) ppm. Spectral data is consistent with literature reports (Rindelaub et al., 2016b).

### 2.3.7.5 Nitrate ester 23

The title compound was isolated as a white, crystalline solid: mp = 132-134 °C; $R_f$ 0.29 (25% EtOAc:Hex; anisaldehyde); IR (ATR) cm[-1]  3294 (bd w, alcoholic OH), 2961 (w), 2360 (w), 1624 (m, $NO_2$), 1282 (m, $NO_2$), 862 (m, $NO_2$); HRMS ESI calc'd for $C_{10}H_{16}O_4N^-$ (214.10797, observed; 214.10738, expected); [1]H NMR (400 MHz, Chloroform-*d*) δ 4.64 (s, 1H), 4.00 (ddd, $J$ = 7.1, 3.3, 0.9 Hz, 1H), 2.37 (ddd, $J$ = 13.8, 7.1, 2.5 Hz, 1H), 1.81 (d, $J$ = 4.3 Hz, 1H), 1.60 – 1.51 (m, 3H), 1.32 (dt, $J$ = 13.8, 4.0 Hz, 1H), 1.23 (s, 3H), 1.22 (s, 3H), 0.89 (s, 3H); [13]C NMR (101 MHz, Chloroform-*d*) δ 94.13, 68.52, 53.01, 47.34, 40.49, 37.76, 37.10, 29.24, 19.12, 14.75 ppm.

### 2.3.7.6 Nitrate ester 24

The title compound was isolated as a white, crystalline solid: mp = 62-64 °C; $R_f$ 0.24 (15% EtOAc:Hex; anisaldehyde); HRMS ESI calc'd for $C_{10}H_{17}O_4NNa^+$ (238.10459, observed; 238.10498, expected); IR (ATR) cm[-1] 3434 (w, alcoholic OH), 2959 (w), 1622 (s, $NO_2$), 1284 (vs, $NO_2$), 856 (m, $NO_2$); [1]H NMR (400 MHz, Chloroform-*d*) δ 5.31 (ddd, $J$ = 9.9, 3.8, 2.1 Hz, 1H), 4.12 (m, 1H), 2.69 – 2.48 (m, 2H), 1.87 (t, $J$ = 4.9 Hz, 1H), 1.43 (dd, $J$ = 14.0, 3.7 Hz, 1H), 1.24 (dd, $J$ = 13.7, 4.7 Hz, 1H), 1.11 (s, 3H), 0.97 (s, 3H), 0.94 (s, 3H); [13]C NMR (101 MHz, Chloroform-*d*) δ 91.04, 78.09, 52.58, 49.53, 42.73, 39.05, 36.71, 20.20, 19.70, 11.79 ppm.





### 2.3.8 Nitration of β-pinene

Nitration of β-pinene oxide (11.1 mmol) was carried out according to the general procedure in toluene. Purification by column chromatography (80 g SiO$_2$, 0-30% EtOAc/Hex over 20 CV) yielded a 1:1 mix of myrtenol (**26**) and nitrate **29** (344 mg, 9 % combined yield), perillic alcohol (**28**, 600 mg, 36% yield) and nitrate

**27** (71 mg, 3% yield). The combined myrtenol and nitrate **29** were separated by column chromatography (40 g SiO$_2$; 5-20% EtOAc/Hex over 25 CV).

### 2.3.8.1 Myrtenol (26)

The title compound was isolated as a clear, colorless oil: R$_f$ 0.25 (15% EtOAc:Hex; anisaldehyde); $^1$H NMR (400 MHz, Chloroform-*d*) δ 5.49 (dt, *J* = 3.0, 1.5 Hz, 1H), 4.00 (t, *J* = 1.8 Hz, 2H), 2.42 (dt, *J* = 8.6, 5.6 Hz,

1H), 2.37 – 2.20 (m, 2H), 2.19 – 2.09 (m, 2H), 1.61 – 1.45 (m, 1H), 1.31 (s, 3H), 1.19 (d, *J* = 8.6 Hz, 1H), 0.85 (s, 3H) ppm. Spectral data is consistent with literature reports (Motherwell et al., 2004)

### 2.3.8.2 Nitrate 27

The title compound was isolated as a clear colorless oil: IR (ATR) cm$^{-1}$ 3362 (bd w, alcoholic OH), 2947 (w), 1624 (s, NO$_2$), 1281 (vs, NO$_2$), 864 (m, NO$_2$); HRMS ESI calc'd for C$_{10}$H$_{16}$O$_4$N$^-$ (214.10813, observed;

214.10738, expected); $^1$H NMR (400 MHz, Chloroform-*d*) δ 5.57 – 5.49 (m, 1H), 3.71 (q, *J* = 11.5, 8.5 Hz, 2H), 2.55 (ddt, *J* = 14.1, 9.8, 3.9 Hz, 1H), 1.95 – 1.83 (m, 2H), 1.76 (t, J = 4.5 Hz, 1H), 1.52 – 1.22 (m, 5H), 1.10 (s, 3H), 1.01 (s, 3H); $^{13}$C NMR (101 MHz, Chloroform-d) δ 85.30, 62.08, 53.45, 48.62, 45.59, 35.78, 27.53, 22.88, 20.20, 20.02 ppm.

### 2.3.8.3 Nitrate 29

The title compound was isolated as a white, crystalline solid: mp = 59-61 °C; R$_f$ 0.55 (15% EtOAc:Hex; anisaldehyde); IR (ATR) cm$^{-1}$ 3374 (bd w, alcoholic OH), 2915 (w), 1625 (s, NO$_2$), 1281 (vs, NO$_2$), 863 (m, NO$_2$); HRMS ESI calc'd for C$_{10}$H$_{17}$O$_4$NNa$^+$ (238.10474, observed; 238.10498, expected); $^1$H NMR (400 MHz, Chloroform-*d*) δ 4.92 (d, *J* = 1.6 Hz, 1H), 3.75 (d, *J* = 11.3 Hz, 1H), 3.65 (d, *J* = 11.3 Hz, 1H), 1.90 – 1.80 (m, 3H), 1.72 – 1.63 (m, 1H), 1.56 – 1.50 (m, 1H), 1.24 – 1.20 (m, 4H), 0.96 (s, 3H); $^{13}$C NMR (101 MHz,

Chloroform-*d*) δ 89.49, 63.82, 54.38, 48.25, 40.28, 37.03, 29.26, 24.74, 22.11, 19.46 ppm.


# 3 Results and discussion

## 3.1 Synthesis

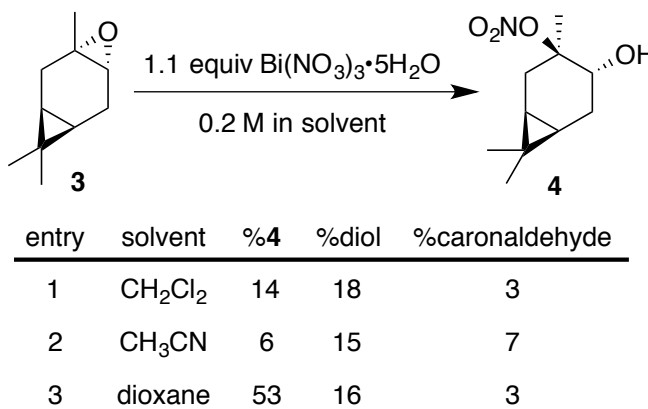

| entry | solvent | %**4** | %diol | %caronaldehyde |
|-------|---------|--------|-------|----------------|
| 1 | $CH_2Cl_2$ | 14 | 18 | 3 |
| 2 | $CH_3CN$ | 6 | 15 | 7 |
| 3 | dioxane | 53 | 16 | 3 |

**Figure 2: Effect of solvent on nitration of *trans*-(3)-carene oxide.**

Previous reports have described the opening of epoxides and aziridines using bismuth nitrate in acetonitrile (Das
et al., 2007). Others report optimal conditions in dichloromethane (Rindelaub et al., 2016b), or 1,4-dioxane with
undesired side reactions in acetonitrile (Pinto et al., 2007). Thus, we began our investigations into the nitration
of *trans*-carene oxide (**3**) using bismuth nitrate in various solvents (Figure 2). In our hands, dioxane clearly
outperformed dichloromethane, yielding 53% of the desired product after 45 minutes (entry 3). A diol was

isolated as a major (16%) byproduct. Attempts to mitigate hydrolysis by the use of base or molecular sieves
were ineffective. Acetonitrile produced a complex intractable mixture with a high amount of diol and
caronaldehyde (entry 2). Other solvents, such as THF and nitromethane, also produced complex mixtures with
no trace of desired nitrate. Methanol, interestingly, only produced an undesired methyl ether. It is unclear if this
product was generated through methanolysis of the nitrate ester or direct substitution of the epoxide. The

regiochemistry of the nitration was easily elucidated with gHSQC NMR data. A doublet of doublets at 3.69 ppm
correlated with a carbon shift at 71.1 ppm and was thus assigned to the methyne adjacent to the alcohol. A
tetrasubstituted carbon at 95.6 ppm was consistent with the tetrasubstituted nitrate ester. Finally, the IR of
compound **4** displayed the expected strong absorbances at 1616, 1291, and 868 cm$^{-1}$.





**Figure 3: Preparation of *cis*-(3)-carene oxide and attempted nitration.**

Next, we aimed to investigate the impact of relative stereochemistry on the nitration. We hypothesized that the secondary nitrate would predominate due to classical stereoelectronic effects. (+)-(3)-Carene was treated with

5 N-bromosuccinimide and calcium carbonate to produce a bromohydrin which subsequently cyclized under basic conditions to produce *cis*-carene oxide (Cocker and Grayson, 1969). Interestingly, we never observed the desired nitrate ester. Instead a 1,2-hydrogen shift generated *cis*-4-caranone in 53% yield.

**Figure 4: Preparation of 8.9-limonene oxide and attempted nitration.**



The preparation of 8,9-limonene oxide (**9**) began with a Diels-Alder reaction to form 1-(4-methyl-3-cyclohexene) ethenone (**8**) in 88% yield. Corey-Chaykovsky addition of a sulfur ylide cleanly produced 8,9-limonene oxide (**9**). Nitration of **9** was attempted in a variety of solvents (DCM, benzene, dioxane, acetonitrile, nitromethane). In acetonitrile, diol **10** was cleanly produced, which rearranged upon purification to aldehyde **11**.

5 In dioxane, aldehyde **11** was observed along with a product consistent with a tertiary nitrate ester. Attempts to purify this mixture on silica gel gave small amounts of **11** and unidentifiable decomposition products. No reaction was observed when non-polar solvents such as DCM and benzene were employed. We hypothesize that the steric hindrance at the β-carbon severely limits reactivity. Furthermore, the fast hydrolysis rates of tertiary nitrate esters hinder the ability to isolate these products.



**Figure 5: Nitration of *cis*- and *trans*-1,2-limonene oxide.**

Commercially available 1,2-limonene oxide was resolved into pure samples of *cis*- and *trans*-1,2-limonene oxide by treating with cyclic amine bases (Steiner et al., 2002). The nitration of *cis*-1,2-limonene oxide (***cis*-12**) proceeded smoothly in both dioxane (53%) and dichloromethane (63%). Correspondingly, the nitration of trans-

15 1,2-limonene oxide (***trans*-12**) also proceeded smoothly in both dioxane (63%) and dichloromethane (61%). The *cis* isomer was predicted to produce the tertiary nitrate ester via stereoelectronic effects. Respectively, the trans isomer was expected to produce the secondary nitrate ester. Contrary to previous reports (Romonosky et al., 2015), we did not observe the desired nitrate esters in acetonitrile. gHSQC NMR data was used to verify nitrate ester **13** was tertiary and **14** was secondary. For **13**, a broad singlet at 4.12 ppm correlated with a carbon shift at

20 69.2 ppm and was thus assigned to the methyne adjacent to the alcohol. A tetrasubstituted carbon at 91.4 ppm was consistent with the tetrasubstituted nitrate ester. Similarly for **14**, a broad triplet at 5.00 ppm correlated with a carbon shift at 84.45 ppm and was thus assigned to the methyne adjacent to the nitrate ester. A tetrasubstituted carbon at 69.5 ppm was consistent with the tertiary alcohol.





**Figure 6: Nitration of perillic alcohol oxide.**

Under acidic conditions, β-pinene undergoes facile rearrangement to form perillic alcohol. Accordingly, we prepared the epoxide **15** by treating perillic alcohol with 1 equivalent of m-CPBA. The inseparable mixture of diastereomers were treated with bismuth nitrate under the standard reaction conditions. Nitrate esters **16** and **17** were easily separated by column chromatography and assigned to the secondary and tertiary nitrate esters, respectively. These isomers were formed from the *cis-* and *trans-*epoxide diastereomers in an analogous fashion to the limonene isomers. For **16**, a broad singlet at 5.21 ppm correlated with a carbon shift at 80.1 ppm and was thus assigned to the methyne adjacent to the nitrate ester. A tetrasubstituted carbon at 70.9 ppm was consistent with the tertiary alcohol. Similarly for **15**, a broad singlet at 4.22 ppm correlated with a carbon shift at 65.6 ppm and was thus assigned to the methyne adjacent to the alcohol. A tetrasubstituted carbon at 92.8 ppm was consistent with the tertiary nitrate ester.




| entry | solvent | temp (°C) | additive | %19 | %20 | %21 | %22 | %23 | %24 |
|---|---|---|---|---|---|---|---|---|---|
| 1 | CH$_2$Cl$_2$ | 23 | -- | 21 | 2 | 5 | 2 | 1 | 4 |
| 2 | CH$_2$Cl$_2$ | -78 | -- | -- | -- | -- | -- | -- | -- |
| 3 | CH$_2$Cl$_2$ | -78 | 1 equiv TBAN | 20 | -- | 6 | 13 | 3 | -- |
| 4 | dioxane | 23 | -- | 25 | -- | -- | 5 | 5 | 2 |
| 5 | Toluene | 23 | -- | 22 | 1 | 4 | 4 | 8 | -- |
| 6 | Toluene | 23 | 1 equiv TBAN | 28 | 8 | 13 | 6 | 4 | 2 |

**Figure 7: Nitration of α-pinene oxide.**

The facile rearrangements of α-pinene with both BrØnsted and Lewis Acids (Kaminska et al., 1992) can be thought to proceed via a non-classical isobornyl cation (**18**) (Kong et al., 2010). We expected the reaction with

5 bismuth nitrate to generate a complex mixture of products due to the many reactive sites. As shown in Figure 7, three rearrangement products (campholenic aldehyde **19**, diene **20**, and *trans*-carveol **21**) and three nitrate esters (**22**, **23**, **24**) were observed. The structure of nitrate ester **23** was identified by a clear singlet at 4.64 ppm (s, 1H). 2D-NMR data (gCOSY, gHSQC and gHMBC) along with comparison to 6-exo-hydroxyfenchol, the analogous diol confirmed the structural assignment (Miyazawa and Miyamoto, 2004). Correspondingly, nitrate

ester **24** displayed a distinct doublet of doublets of doublets at 5.31 ppm that correlated to the methyne adjacent to the nitrate ester. Again, 2D NMR data (gCOSY, gHSQC and gHMBC) was used in conjunction with the literature spectra for platydiol and its epimer to verify the assignment (KUO et al., 1989).

Under all conditions campholenic aldehyde was the major product (20-28% yield). First generation nitrate ester

**25** was not found under all conditions. All six components were isolated in small amounts when the reaction was run in dichloromethane (Figure 7, entry 1). Cooling the reaction to -78 °C completely shut down all



reactivity (entry 2). Interestingly, adding one equivalent of tetrabutylammonium nitrate (TBAN) as an external nitrate source in addition to bismuth nitrate at -78 ºC generated mostly nitrate **22** (entry 3). More polar solvents, like dioxane (entry 4), generated slightly higher amounts of nitrates at room temperature. Aromatic solvents, such as benzene and toluene provided the best yield of **23.** Interestingly, adding TBAN to the nitration in

5    toluene dramatically increased the amounts of diene **20** and *trans*-carveol **21**. Using acetonitrile as a solvent generated a complex mixture that appeared to be mainly diols. Zirconium nitrate produced a similar distribution of products (Das et al., 2006), but other metal nitrate complexes (Y(NO$_3$)$_3$•6H$_2$O, Co(NO$_3$)$_2$•6H$_2$O) resulted in no observable reaction.

**Figure 8: Nitration of β-pinene oxide.**

The nitration of β-pinene was similarly complicated by isomerization pathways through the corresponding non-classical carbocation. Nitrate **27** initially co-eluted with myrtenol (**26**) but was separable upon a second purification by column chromatography. The methyne proton of **27** at 5.51 ppm is correlated with a carbon at 85.3 ppm in the gHSQC NMR. This data is consistent with the secondary nitrate ester. Similarly, the methyne

15    proton in nitrate **29** is a clear singlet at 4.91 ppm. This signal correlates with a carbon at 89.5 ppm. Interestingly, nitrates **30** and **31** were not observed.

**3.2 Spectral data**

| compound | C**H**ONO$_2$ (ppm) | C**H**ONO$_2$ (ppm) | IR (cm$^{-1}$) | | |
|---|---|---|---|---|---|
| **4** | -- | 95.6 | 1616 | 1291 | 868 |



| Structure | | | | | |
|---|---|---|---|---|---|
| **13** | -- | 91.4 | 1618 | 1291 | 860 |
| **14** | 5.00 (s, 1H) | 84.4 | 1627 | 1279 | 876 |
| **16** | 5.21 (s, 1H) | 80.1 | 1627 | 1280 | 880 |
| **17** | -- | 92.5 | 1623 | 1288 | 863 |
| **22** | -- | 94.2 | 1614 | 1293 | 868 |
| **23** | 4.64 (s, 1H) | 94.13 | 1624 | 1282 | 862 |
| **24** | 5.31 (ddd, J = 9.9, 3.8, 2.1 Hz, 1H) | 91.04 | 1622 | 1284 | 856 |
| **27** | 5.51 (m, 1H) | 85.3 | 1624 | 1281 | 864 |
| **29** | 4.91 (s, 1H) | 89.5 | 1625 | 1281 | 863 |

**Table 1. Collated $^{1}$H and $^{13}$C NMR data with IR absorbances for nitrate esters.**

As shown in table 1, all nitrate esters displayed the expected strong nitrate ester absorbances at ~1600, 1300, and 900 cm$^{-1}$. Methyne protons next to the secondary nitrate esters appeared between 4.64 ppm and 5.51 ppm. Carbon chemical shifts were found over a broader range than anticipated from 80.1 to 95.6 ppm. Finally, we observed masses consistent with methanolysis products (185.1 m/z M+; 223.1300 m/z M+Na) when nitrate esters were analysed (GC-MS and HRMS) as solutions in methanol. Further experimentation is necessary to evaluate the implications of methanolysis.



### 3.3 Stability and storage

We were surprised to find that five of the nitrate esters (**4**, **16**, **23**, **24**, and **29**) are solids. We have stored these compounds at 0 ºC for up to 9 months with no noticeable decline in purity. The remaining nitrate esters are stable at 0 ºC for 2-4 weeks. Interestingly, some of these compounds appear to deviate from the expected

stability patterns (*vide supra*). For example, tertiary nitrate ester **4** is particularly stable, whereas secondary nitrate ester **14** decomposed within a few weeks at 0 ºC.  Freezing the samples as a solution in benzene is recommended for longer term (~6 month) storage.

### 4 Conclusions

We have clearly delineated the methods required to synthesize and purify nine nitrate esters derived from mono-

terpenes. Seven of these compounds are undescribed in the literature and the remaining two had gaps in their characterization. We believe that the availability of these compounds will enable further study of the structure-reactivity relationships. For example, comparing the specific hydrolysis rates for tertiary versus secondary nitrate esters in **13** and **14** as well as **17** and **16** could help deconvolute the fates of each terpene in the atmosphere. We also believe that these compounds will assist in confirming the identities of organic nitrates that

have previously been limited to detection by MS based methods (Rindelaub 2016b). Using our methods it is possible to cleanly isolate 50-100 mg even of the least prevalent isomers from the nitration reactions of α-pinene and β-pinene. The availability of these compounds is important further studies into the influence of terpene structure on the fate and roles of organic nitrates in SOA formation.

**Supplementary material related to this article is available online at:**

*Author contributions.* RL designed the experiments and prepared the manuscript with contributions from all co-authors. All co-authors carried out experiments and contributed to reviewing and editing the manuscript.

*Acknowledgements.* This research was supported by generous start-up funding from Reed College. E.A.M. is grateful for funding from the Clark Fellowship. We thank Julie Fry for drawing our attention to this class of

compounds. We are particularly grateful to Drew Gingerich, Nick Till, Stewart Green, Alyssa Harrison, Carlo Berti for preliminary experiments and to the 2014 and 2015 Chem 343 students for their assistance in this research.



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
