# Peer review of "Technical Note: Preparation and purification of atmospherically relevant $\alpha$ -hydroxynitrate esters of monoterpenes"

_Atmospheric Chemistry and Physics, 2019_

## Referee Comment (RC1) · Anonymous Referee #1 · 14 Aug 2019

General comments

This manuscript describes the synthesis and characterization of a number of hydroxy nitrate esters of monoterpene compounds. Specifically, the work refines a previously identified route to hydroxy nitrate esters – reaction of epoxide precursors with bismuth nitrate – and shows that it is a general pathway to a variety of potentially important nitrate esters. Because of the need for analytical standards for organic nitrates, this work will be of interest to readers of Atmospheric Chemistry and Physics. I recommend publication after the authors consider the following suggestions for improvement of the manuscript.

[Figure]

Specific comments

p. 2 line 24: The cited work was not controversial - it was a case of scientific fraud. I would change this last clause to "and has been subject of a retracted study."

p. 3 line 7: It seems as if the column chromatography method is critical to the success of this work (intractable mixtures turn out to be tractable). Is this particular system critical to this success? I'm not familiar with it.

p. 3 line 25: 13C NMR data is very useful when using reference data to analyze complex mixtures. I see that some species have 13C characterization, and some don't (like this species). It would strengthen the utility of this work if as much 13C characterization were included.

p. 7 line 1: There should be a reference to the numbering system here (or preceding this section) and the information given that the structures are given later in the paper. I also think the name should include monoterpene precursor to help keep track of the various species.

p. 19 line 18: The authors should expand on how their results bear on previous work. Generally, it would be valuable to point out that these reactions generally lead to ring opening of the bicyclic backbone of the monoterpene. The atmospheric chemistry literature contains a number of proposed monoterpene epoxide reaction products in which the bicyclic backbone is retained (for example, see Duporte, G et al. Experimental Study of the Formation of Organosulfates from $\alpha$-Pinene Oxidation. Part I: Product Identification, Formation Mechanisms and Effect of Relative Humidity. The Journal of Physical Chemistry A 2016, 120 (40), 7909–7923) even though the synthetic literature shows that this is not usually the case. Specifically, it should be pointed out that Rindelaub's compounds 1 and 2 are the present work's compounds 17 and 22, and some comparison of the two methods should be given. The authors should also summarize the species for which they were not able to isolate compounds (such as several tertiary nitrates, which have previously been shown to highly susceptible to hydrolysis) since

this is valuable information concerning the potential stability of these compounds in the atmosphere.

Technical corrections

p. 2 line 10: typo (incomplete citation)

p. 11 line 1: type: add "oxide" to heading name

Figure 7, Compounds 23 and 24: It is very difficult to see the dashed line bonds to the nitrate groups.

p. 16 line 3: typo: slashed "o" should be lower case

p. 16 line 12: typo: author name is currently in all capital letters
* * *

---

## Referee Comment (RC2) · Anonymous Referee #2 · 15 Aug 2019

The comment was uploaded in the form of a supplement:
https://www.atmos-chem-phys-discuss.net/acp-2019-690/acp-2019-690-RC2-supplement.pdf

---

## Referee Comment (RC3) · Anonymous Referee #3 · 19 Aug 2019

Note: it appears that superscripts, subscripts, italics and bold type are not recognized by the software. It would be helpful if the editors could provide some means of uploading comments as attachments.

This review is submitted with the benefit of having read reviews of anonymous Reviewers 1 and 2. As stated in the previous reviews, this work contributes procedures for synthesis of a number of vicinal hydroxynitrate esters derived from a monoterpene. The esters can serve as much needed authentic standards for characterizing components of SOA and therefore the work should be published. There are a number of comments that the authors need to address prior to publication.

[Figure]

Most important: the authors should include 1H and 13C NMR traces for all compounds that have not been previously reported, and should also include HSQC and HMBC spectra where this is critical for structural elucidation. In particular, the assignment of quaternary carbons is readily evident from the absence of C–H cross peaks in the HSQC spectra.

Since the authors had access to high-resolution mass spectrometry, exact mass measurements and fragmentation patterns should be provided either in the experimental section of the text or the SI for all compounds not previously reported.

Globally, there is considerable carelessness with regard to the presentation of this manuscript. Conventionally, solvent names do not begin with upper case letters and in the case of deuterated solvents, the notation "d" is italicized (e.g., chloroform-[italics]d). This convention is adhered to randomly throughout.

The use of abbreviations for solvents is also not consistent (e.g. EtOAc:Hex and ethyl acetate:hexanes (solvent and compound names are inappropriately capitalized in the text)).

Several citations are only partially given in the text (e.g. Cocker, 1969 on p. 4, l. 8) and in one case, a citation in the text is missing from the bibliography (Charbonneau, 2018; Charbonneau was misspelled).

There are mistakes in nomenclature: trans-3-carene (not trans-(3)-carene). Synthetic targets are referred to as "title" compounds. However, "target compound" would be preferable, since compounds do not appear in titles.

Reaction/synthetic schemes should be separated from tables and figures and appropriately designated. Although the comments above are editorial and easy to correct, the resulting impression is a lack of care in preparation of the manuscript.

There are also some specific issues that need to be addressed: 1. The authors might note that products having two or more asymmetric carbons were isolated and charac-

terized as mixtures of diastereomers. Visual inspection of the NMR spectra does not appear to how any resolution of the mixture by NMR. Did the authors look for any such result?

2. For the preparation of trans-carene oxide (p. 3, starting with l. 17), did the authors start with optically pure (+)-carene or a racemic mixture? This is important, since epoxidation with mCPBA will give an enantiomerically pure epoxide from enantiomerically pure starting material. The synthesis of the cis oxide described below apparently starts with (+)-carene.

3. Preferred nomenclature for cpd 20 would be 2-methyl-5-(propan-2-ylidine)cyclohex-2-en-1-ol.

4. Visual inspection of the 1H NMR trace of cpd 24 appears to be consistent with the signal at 5.31 ppm as a poorly resolved doublet-of-doublets. It is not clear how the authors interpreted this signal as a doublet-of-doublets-of-doublets.

5. p. 7, l. 16 and following; p. 13, l. 3 and following: As pointed out by Reviewer 1, there is a problem with the structural assignment of limonene 8,9-dihydrodiol (10) to the crude product of Bi(NO3)3 treatment of limonene 8,9-oxide (9). This is true as well for the rearrangement product 2-(4-methylcyclohex-3-en-1-yl)propanal (preferred nomenclature) (11). The 1H NMR data provided do not match published data for either proposed structure (diol: J. Am. Chem. Soc. 2018, 140, 1502−1507; aldehyde: Phytochemistry 2017, 144, 208 − 215). This reviewer notes that the reported 1H NMR of the diol was acquired at a spectrometer frequency of 600 MHz, although the only expected difference would be improved resolution of the proton splittings.

6. On p. 11, starting with line 12: The authors should provide additional discussion justifying the structural assignments of cpds 27 and 29. Neither of these compounds appears to have been structurally characterized by NMR in the literature. In the 1H NMR spectrum of 27, the signal at ∼5.6 ppm appears to be a broad doublet. It is unclear why the authors have used the designation "multiplet". In the 1H NMR spectrum of 29,

which is a racemic mixture, the lowest field signal (H attached to the nitrate-substituted C) should be a singlet, as it appears to be. Why is it described as a doublet?

7. Figure 2: The authors should add the diol and aldehyde structures to this scheme. There is sufficient space and this gives the true picture of reaction products.

8. p. 12. l. 15: The phrase, "...assigned to the methyne adjacent to the alcohol..." should be rewritten as "...assigned to the alcohol methyne carbon..." As written, the statement "adjacent to" is confusing. Where heteronuclear correlations are important in verifying structure, the 2D spectra should be added to the SI. As pointed out above, it would be helpful if all the NMR data were included in SI.

9. p. 12, l. 17: The rationale for assigning the quaternary carbon would be better explained by stating that no C–H cross peak was observed for the 13C signal at 95.6 ppm in the HSQC experiment.

10. p. 16, l. 12: "Epimer" is misused here. Platydiol (the name "platydiol" automatically designates the the di-endo geometry of the diol) has a plane of symmetry, and there are no optical isomers.

11. p.17, l. 14: In discussing the structural verification of nitrate ester 27, it would be helpful to include the HSQC spectrum. As presented in the text, there is not enough data to be convincing regarding the structural assignment.

12. p. 17, bottom: The table of spectral data should have the heading at the top.

Please also note the supplement to this comment:
https://www.atmos-chem-phys-discuss.net/acp-2019-690/acp-2019-690-RC3-supplement.pdf

───────────────────────

---

## Referee Comment (RC4) · Anonymous Referee #4 · 31 Aug 2019

General Comments:

The manuscript presents synthesis methods for nitrate esters with relevance to terpene oxidation products in the atmosphere. The authors report preparation and purification of nine nitrate esters and additionally provide some details on unsuccessful methods. The information contained in this manuscript will be of interest to atmospheric chemists looking into nitrate radical oxidation products and I recommend acceptance after the following comments are addressed.

Specific comments:

1. Was the HPLC listed on page 3 line 12 used on the Orbitrap? If so, what conditions

were used (solvents, gradient, spray conditions, etc.). Were both positive and negative ion mode used? How was the instrument mass calibrated?

2. Acronyms I noticed that should be added to the list: CV, TLC, br, dq and qd, vs, Rf etc.

3. Exact mass measurements were only provided for a subset of the samples. Why is this the case? Also, the expected exact masses for the anions are incorrect, check to make sure the mass of the electron has been added. The expected exact mass for C10H16O4N- is 214.10848. (page 9 line 6 and page 10 line 16).

4. Methanolysis products were observed when methanol was used as the analysis solvent (page 18). Were these the only products observed or were the desired products also observed? Did this occur for all the nitrate esters? Also, please clarify what M+ is for each of the masses. I can see how m/z 185.1 is formed but I do not see where m/z 223.13 is coming from. Also, are these exact mass measurements? If so, providing more numbers after the decimal is a good idea.

5. Figure 7 is very crowded, especially at the far right side on top. I recommend providing a little more space between compounds 22, 23 and 24 so that they are easier to see.

Technical notes:

1. Incomplete citation on page 2, line 10

2. m/z should be italic throughout

3. Formatting on the citations looks a little odd, especially the spacing between the comma and the dates

---

## Referee Comment (RC5) · Anonymous Referee #5 · 9 Sep 2019

The paper by McKnight et al. presents successful synthetic routes to the preparation of the hydroxy nitrates that are produced in nature by the atmospheric oxidation of monoterpenes. In recent years it has been shown that formation of these organic nitrates are an important sink for NOx in the atmosphere, as well as an important mechanism for the production of atmospheric aerosol. Such aerosol results from an interaction between anthropogenic NOx, and biogenic monoterpenes. Thus, interestingly, control of NOx emissions (from combustion) can influence the role of production of atmospheric aerosol from biogenic emissions. There is thus considerable interest in these compounds, and their atmospheric chemistry, and, e.g. the ability to study their hydrolysis rates. However, the atmospheric chemistry community has not had much

success in making these compounds, and that is a major stumbling block to progress. Here is a paper that connects the organic synthesis community to the atmospheric chemistry community in a useful way, and the atmospheric chemistry community increasingly needs this connection, in more general terms. This paper is technically important, and will stimulate a great deal of badly needed research, likely to start with atmospheric chemists at Reed. So, this paper should be published, and I think can be done with only editorial changes. That said, the authors need to be aware that the audience here is indeed atmospheric chemists, who are not versed in the editorial jargon of synthetic organic chemists. So, the paper could be made more readable and useful if the authors shed the notion that they are writing to organic chemists. They are, at best, writing to organic chemists existing with the labs of atmospheric chemistry professors.

Comments and suggested changes are listed below, in the order they arose in the manuscript.

1. The last sentence of the Introduction is unclear – 30-40% of MT emissions are at night, or the nitrate radical consumes 30-40% of the emitted MTs at night? Please note that the OH radical oxidation pathway is still an important source of MT-nitrates, e.g. as discussed in Pratt et al., 2012.

2. Line 21, page 2 – is there a better reference for this quite general organic chemistry laboratory hazard?

3. The Ma et al. paper should not be cited. If you go to the ACP web site, it says "This paper has been retracted."

4. Page 3, line 17 – what is CPBA? (Good example of my comment above; most organic chemists will know that it doesn't stand for Certified Professional Behavioral Analyst).

5. Page 3, line 27 – should the reference be Crocker and Grayson, 1969? And on Page

4, line 8.

6. Page 5, line 8 should be Steiner et al., 2002. And line 16.

7. Page 7, at 2.3.1 title, refer the Table 1 for structures. The reader needs to see the structures. Line 2 – do you mean the trans-3-carene epoxide? Also on line 10, the epoxide?

8. Page 8, line 4, those percentages are the yields? Please try to avoid organic synthetic chemists shorthand. Also line 13. For structures 16 and 17, line 22, are there yields?

9. Page 8, line 24, what is "the title compound"? Also page 9, lines 4, 17, and 23. There is sloppy shorthand on this page, e.g. line 13 – "the following". What following?

10. Line 11 – you mean the a-pinene oxide? Line 17 – by "title compound" you mean the aldehyde? Similarly, I find line 23 confusing. You synthesized the alcohol?

11. Page 10, lines 13 and 14 – "The title compound", when the title compound is "Nitrate ester 23" does not make for great/easy reading. Perhaps it would be helpful that in each case, the structure should be shown next to the name, as a graphic? That would really make the paper more readable.

12. Page 14, line 8 – provide a reference for the fast hydrolysis rates.

13. Page 19, line 6, re decomposition of ester 14 – does this depend on water content in the solvent? Do you know what the decomposition products are? The corresponding diol?

14. Page 19, line 9 – the "the" methods, but "successful methods".

15. Line 17 – insert "for" after "important".

---

## Author Comment (AC1) · 12 Nov 2019

**Response to Referee #1**

Our answers are in bullet points below the original referee comments in **bold**, changes to the manuscript are in *italics*.

**This manuscript describes the synthesis and characterization of a number of hydroxy nitrate esters of monoterpene compounds. Specifically, the work refines a previously identified route to hydroxy nitrate esters – reaction of epoxide precursors with bismuth nitrate – and shows that it is a general pathway to a variety of potentially important nitrate esters. Because of the need for analytical standards for organic nitrates, this work will be of interest to readers of Atmospheric Chemistry and Physics. I recommend publication after the authors consider the following suggestions for improvement of the manuscript.**

Thank you for the acknowledgement of the importance of our work.

1. **p. 2 line 24: The cited work was not controversial - it was a case of scientific fraud. I would change this last clause to "and has been subject of a retracted study."**
   - We were unaware of the specific circumstances of the retraction and have corrected the last sentence to read *"and has been subject of a retracted study."*
2. **p. 3 line 7: It seems as if the column chromatography method is critical to the success of this work (intractable mixtures turn out to be tractable). Is this particular system critical to this success? I'm not familiar with it.**
   - Automated chromatography systems have become common in both academic and industrial organic laboratories. Teldyne Isco and Biotage are two of the major companies that manufacture such systems. Typically the chromatography systems can handle higher pressures and also finer grades of silica. The increased pressure, finer grades of silica, coupled with the accuracy of pumps for gradient elutions leads to better separations than standard glass column chromatography. We have not tested the purification of these compounds by methods other than the CombiFlash Isco. We have tested other lower grades (40-63 μm irregular) of silica and have found that the finer grade (25 μm spherical silica) gives optimal separation.
3. **p. 3 line 25: 13C NMR data is very useful when using reference data to analyze complex mixtures. I see that some species have 13C characterization, and some don't (like this species). It would strengthen the utility of this work if as much 13C characterization were included.**
   - We have included full characterization ($^1$H & $^{13}$C NMR, IR, HRMS, melting point, $R_f$) for all new compounds. Known compounds, such as *trans*-carene oxide, have been described by their $^1$H NMR and a literature reference, as is standard practice.
4. **p. 7 line 1: There should be a reference to the numbering system here (or preceding this section) and the information given that the structures are given later in the paper. I also think the name should include monoterpene precursor to help keep track of the various species.**
   - We have appended structures for each experimental procedure to clarify the numbers and structures.
5. **p. 19 line 18: The authors should expand on how their results bear on previous work. Generally, it would be valuable to point out that these reactions generally lead to ring**

**opening of the bicyclic backbone of the monoterpene. The atmospheric chemistry literature contains a number of proposed monoterpene epoxide reaction products in which the bicyclic backbone is retained (for example, see Duporte, G et al. Experimental Study of the Formation of Organosulfates from α-Pinene Oxidation. Part I: Product Identification, Formation Mechanisms and Effect of Relative Humidity. The Journal of Physical Chemistry A 2016, 120 (40), 7909–7923) even though the synthetic literature shows that this is not usually the case. Specifically, it should be pointed out that Rindelaub's compounds 1 and 2 are the present work's compounds 17 and 22, and some comparison of the two methods should be given. The authors should also summarize the species for which they were not able to isolate compounds (such as several tertiary nitrates, which have previously been shown to highly susceptible to hydrolysis) since this is valuable information concerning the potential stability of these compounds in the atmosphere.**

- We have added the following to the conclusions section "*Interestingly, we did not observe the formation of a-pinene oxide and b-pinene oxide products that retained their bicyclic ring structures. This is consistent with the solution phase synthetic literature (Kaminska et al., 1992) but in contrast to many atmospheric reports (For example, see: Rindelaub, 2016b; Duporte, G., et al., 2016)."*
- The reports of compounds 1 & 2 lacked reference to relative stereochemistry. Due to the potential presence of diastereomers, we prefer to keep our compounds (that are single diastereomers and indicated relative stereochemistry) labeled separately.

6. **p. 2 line 10: typo (incomplete citation)**
   - citation corrected to *(Nozière et al., 2015)*
7. **p. 11 line 1: type: add "oxide" to heading name**
   - name corrected to *"β-pinene oxide"*
8. **Figure 7, Compounds 23 and 24: It is very difficult to see the dashed line bonds to the nitrate groups.**
   - Decreased the hatch spacing/increased the number of hatches for dashed bonds
9. **p. 16 line 3: typo: slashed "o" should be lower case**
   - corrected to "Brønsted"
10. **p. 16 line 12: typo: author name is currently in all capital letters**
    - corrected capitalization of author names in reference.

---

## Author Comment (AC2) · 12 Nov 2019

**Response to Referee #2**
Thank you for the valuable comments and suggestions. Our answers are in bullet points below the original referee comments in **bold**, changes to the manuscript are in *italics*.

**In the manuscript titled "Preparation and purification of atmospherically relevant a-hydroxynitrate esters of monoterpenes", McKnight et al described the synthesis of a series of nitrate esters. Currently there is large uncertainties in secondary organic aerosol formation especially from organic nitrate esters derived from biogenic volatile organic compounds, in part hindered by the availability of standards. Thus, availability of nitrate ester standard would bridge the critical knowledge gap exists in our further understanding of the mechanism of aerosol formation. This would be of interest to the reader of ACP.**

Thank you for this positive assessment. We also hope that the availability of these compounds will help fill this critical knowledge gap.

1. **First of all, since this is a journal on atmospheric science and the title stated "Preparation and purification of atmospherically relevant a-hydroxynitrate esters of monoterpenes", the reviewer feels that it is worthwhile to more specifically spell out the relevance of the synthetic targets with atmospheric chemistry mechanistically. The way the authors frame the whole set of compounds as a general class "nitrate ester of monoterpenes" is good in a general way with the synthesis of variously nitrate esters as the general aim, a little bit more detailed account on the relevance of all the targeted compounds with atmospheric chemistry would be appropriate for majority of the reader of this journal. The author did an extensive investigation on the reaction of nitrate with epoxide, if that were to be emphasized for its relevance, it would be better to spell out clearly.**
   - We have clearly delineated the importance of these compounds in terms of atmospheric chemistry in the introduction section. As a technical note, we believe that the introduction is sufficient background information to convey the importance of our work and will leave descriptions of the detailed and complicated mechanistic fates to the references sited therein.
2. **As the author presented in the manuscript, IR would provide characteristic peaks for NO2 group, supporting the existence of nitro group in the analyzed molecule. As the indicated in table 1 of this manuscript, compared to the corresponding alcohols, the nitrate esters likely would have very similar NMR spectral profile with only the proton and carbon at the alpha position to the nitrate ester group to likely exhibit significant chemical shift difference. On the other hand, mass spectroscopy would be very important in such characterization if a good spectrum is feasible. The absence of such data from many nitrate ester target compounds in this manuscript would be better to be explained since only the HRMS of a few of them are provided.**
   - HRMS data have been added for all previously unreported nitrate esters.
3. **The opening of the cis and trans-1,2-limonene oxide with Bi(NO2)3 under the same conditions led to different nitrate ester with nitro group attached to different positions to afford different constitutional isomer, instead of stereoisomer. The reason behind it might be the same as the kinetic resolution of the cis and trans-1,2-limonene oxide. This fact merits more discussion and more detailed 2-D NMR spectroscopy evidence in**

**addition to those original 1-D NMR spectrum provided by the author would be very helpful to go with the description text.**

- Added clarifying text and updated figure 5. *"The nitration of the cis isomer was predicted to produce the tertiary nitrate ester via stereoelectronic effects. The dominant conformation of cis-1,2-limonene oxide is expected to be a pseudo-half chair (Figure 5). Nucelophilic substitution at the less substituted position would proceed through a lower energy chair-like transition state, whereas attack at the more substituted carbon leads to a very unstable twist-boat. The nitration of cis-1,2-limonene oxide (**cis-12**) proceeded smoothly in both dioxane (53%) and dichloromethane (63%). Respectively, the trans isomer was expected to produce the secondary nitrate ester."*
- HSQC data have been added to the supplement document

**4. The characterization of limonene-diol doesn't seem to match with other report (J. Am. Chem. Soc. 2018, 140, 1502−1507). Since the other report provided the original NMR spectrum, it would be feasible for the author to provide an explanation of the discrepancy. Furthermore, in figure 4, the author attributed the transformation of compound 10 to 11 and back an interchange equilibrium. That interpretation is better to be explained further since it is not obvious.**

- Thank you for bringing this to our attention. We revisited the reaction of **9** in acetonitrile & dioxane. Upon further characterization ($^{13}$C NMR, gHSQC) we determined that the product formed in acetonitrile is actually a 1:1 diastereomeric mixture of oxazolidine **10a**. Of note were $^{13}$C signals at: 162.62, 162.55 ppm (carbonyl carbon), 72.77, 72.62 ppm (tetrasubstituted C), 13.87, 13.86 ppm ($\underline{C}$H$_3$C). The structure was further confirmed by an IR stretch at 1674 cm$^{-1}$ and an *m/z* 193.8 (M+). This type of reactivity with bismuth nitrate and acetonitrile opening epoxides was also observed by Pinto, et al., 2007.
- The interchange of either limonene-diol or nitrate **10** can be rationalized by an intramolecular elimination and tautomerization.

**5. There are other suggestions**

**Page 1 Line 21. The abbreviation "ON" is not necessary as it is used only once.**

- Amended the text to utilize the abbreviation for organonitrate.

**Page 2 Figure 1. Compound 1 is basically compound 13 without stereo isomer indication**.

- The reports of compounds 1 & 2 lacked reference to relative stereochemistry. Due to the possible presence of two diastereomers, we prefer to keep our compounds (that are single diastereomers and indicated relative stereochemistry) labeled separately.

**Page 3 Line 17. "0.86 mL, 1 g" incorrect.**

- Corrected to "0.86 g, 1 mL"

**Page 3 Line 25. Citation year incorrect.**

- Corrected to 2009

**Page 3 Line 27. Citation better to be placed at the first mention.**

- Amended to "Crocker and Grayson, 1969" at first mention.

*Page 4 line 10. "(9 am – 11 am)" relevant?*

- deleted

*Page 5 line 7. Citation better to be placed at the first mention*

- Amended to "Steiner, et al., 2002" at first mention.

---

## Author Comment (AC3) · 12 Nov 2019

**Response to Referee #3**
Thank you for the valuable comments and suggestions. Our answers are in bullet points below the original referee comments in **bold**, changes to the manuscript are in *italics*.

**This review is submitted with the benefit of having read reviews of anonymous Re- viewers 1 and 2. As stated in the previous reviews, this work contributes procedures for synthesis of a number of vicinal hydroxynitrate esters derived from a monoterpene. The esters can serve as much needed authentic standards for characterizing components of SOA and therefore the work should be published. There are a number of comments that the authors need to address prior to publication.**

Thank you for acknowledging the critical need for these standards. We hope that our work will help fill this gap in the literature.

**Most important: the authors should include 1H and 13C NMR traces for all compounds that have not been previously reported, and should also include HSQC and HMBC spectra where this is critical for structural elucidation. In particular, the assignment of quaternary carbons is readily evident from the absence of C–H cross peaks in the HSQC spectra. Since the authors had access to high-resolution mass spectrometry, exact mass mea- surements and fragmentation patterns should be provided either in the experimental section of the text or the SI for all compounds not previously reported.**

- We have included all $^1$H, $^{13}$C NMR and HSQC spectra in the SI for all compounds not previously reported. We have also added HRMS data in the experimental section for all compounds not previously reported.

**Globally, there is considerable carelessness with regard to the presentation of this manuscript. Conventionally, solvent names do not begin with upper case letters and in the case of deuterated solvents, the notation "d" is italicized (e.g., chloroform-[italics]d). This convention is adhered to randomly throughout.**
**The use of abbreviations for solvents is also not consistent (e.g. EtOAc:Hex and ethyl acetate:hexanes (solvent and compound names are inappropriately capitalized in the text)).**

- We have fixed the minor typos and inconsistent abbreviations.

**Several citations are only partially given in the text (e.g. Cocker, 1969 on p. 4, l. 8) and in one case, a citation in the text is missing from the bibliography (Charbonneau, 2018; Charbonneau was misspelled).**

- We have amended the partial citations to full citations at first instance.
- Charbonneau, 2018 has been added to the bibliography.

**There are mistakes in nomenclature: trans-3-carene (not trans-(3)-carene). Synthetic targets are referred to as "title" compounds. However, "target compound" would be preferable, since compounds do not appear in titles.**

- Corrected instances of extraneous parentheses.
- Replaced "title compound" with the names compounds and compound numbers.

**Reaction/synthetic schemes should be separated from tables and figures and appropriately designated.**
- Sometimes it may be appropriate to separate a table from a reaction scheme, we do not believe that it is appropriate in this case. The tables are communicating different reaction conditions of the connected reaction scheme and would be meaningless if separated.

1. **The authors might note that products having two or more asymmetric carbons were isolated and characterized as mixtures of diastereomers. Visual inspection of the NMR spectra does not appear to how any resolution of the mixture by NMR. Did the authors look for any such result?**
- We have indicated in the text and by using line bonds (not wedged or dashed) the cases where we generated mixtures of diastereomers. For example, the perillic alcohol epoxides **15**, 8,9-limonene oxide (**9**), nitrate ester **10**, oxazoline **10a**, and aldehyde **11** are the only mixtures of diastereomers. All other compounds are drawn, characterized and reported as single diastereomers.
2. **For the preparation of trans-carene oxide (p. 3, starting with l. 17), did the authors start with optically pure (+)-carene or a racemic mixture? This is important, since epox-idation with mCPBA will give an enantiomerically pure epoxide from enantiomerically pure starting material. The synthesis of the cis oxide described below apparently starts with (+)-carene.**
   - The starting material was corrected to (+)-carene.
3. **Preferred nomenclature for cpd 20 would be 2-methyl-5-(propan-2-ylidine)cyclohex- 2-en-1-ol.**
   - Amended to "5-isopropylidene-2-methyl-2-cyclohexen-1-ol"
4. **Visual inspection of the 1H NMR trace of cpd 24 appears to be consistent with the signal at 5.31 ppm as a poorly resolved doublet-of-doublets. It is not clear how the authors interpreted this signal as a doublet-of-doublets-of-doublets.**
   - The expansion of the NMR trace clearly shows the reported doublet-of-doublets-of-doublets. This is corroborated by gCOSY correlation to the adjacent methylene as well as long-range, 4-bond coupling to the methyne (-C$\underline{H}$OH). This type of 4-bond coupling ($J =$ 1-2 Hz) is quite common in substituted cyclohexane derivatives (see: Crews, P., et al. *Organic Structure Analysis,* Oxford University Press, 1998, pages 142-143).
5. **5. p. 7, l. 16 and following; p. 13, l. 3 and following: As pointed out by Reviewer 1, there is a problem with the structural assignment of limonene 8,9-dihydrodiol (10) to the crude product of Bi(NO3)3 treatment of limonene 8,9-oxide (9). This is true as well for the rearrangement product 2-(4-methylcyclohex-3-en-1-yl)propanal (preferred nomenclature) (11). The 1H NMR data provided do not match published data for either proposed structure (diol: J. Am. Chem. Soc. 2018, 140, 1502−1507; aldehyde: Phytochemistry 2017, 144, 208 – 215). This reviewer notes that the reported 1H NMR of the diol was acquired at a spectrometer frequency of 600 MHz, although the only expected difference would be improved resolution of the proton splittings.**
   - Thank you for bringing this to our attention. We revisited the reaction of **9** in acetonitrile & dioxane. Upon full characterization we determined that the product formed in acetonitrile is actually a 1:1 diastereomeric mixture of oxazolidine **10a**. Of note were $^{13}$C signals at: 162.62, 162.55 ppm (carbonyl carbon), 72.77, 72.62 ppm (tetrasubstituted C), 13.87, 13.86 ppm ($\underline{C}$H$_3$C). The structure was further confirmed by an IR stretch at 1674

cm$^{-1}$ and an *m/z* 193.8 (M+). This type of reactivity with bismuth nitrate and acetonitrile opening epoxides was also observed by Pinto, et al., 2007.

- We have added tabulated $^1$H and $^{13}$C NMR for aldehyde 11 in the supplemental information to show the comparison to both Uehara, et al., 2017 and a second reference, Reid and Watson, 2018. The data are in agreement within expected error.

6. **On p. 11, starting with line 12: The authors should provide additional discussion justifying the structural assignments of cpds 27 and 29. Neither of these compounds appears to have been structurally characterized by NMR in the literature. In the 1H NMR spectrum of 27, the signal at ~5.6 ppm appears to be a broad doublet. It is unclear why the authors have used the designation "multiplet". In the 1H NMR spectrum of 29, which is a racemic mixture, the lowest field signal (H attached to the nitrate-substituted C) should be a singlet, as it appears to be. Why is it described as a doublet?**

- The structural assignments of **27** and **29** are in the body of the text (after figure 8) and bear distinct similarities to the related nitrate esters **23** and **24**. The signal at 5.6 appears as a poorly resolved doublet-of-doublet-of doublets, i.e. multiplet. As with **24**, this is due to long-range, 4-bond coupling.
- For **29**, the $^1$H NMR signal is a doublet with the reported 1.6 Hz coupling constant. Again, this is due to apparent long-range, 4-bond coupling and is corroborated by gCOSY data.

7. **Figure 2: The authors should add the diol and aldehyde structures to this scheme. There is sufficient space and this gives the true picture of reaction products.**

- Included diol and aldehyde structures in figure 2.

8. **p. 12. l. 15: The phrase, ". . .assigned to the methyne adjacent to the alcohol..." should be rewritten as ". . .assigned to the alcohol methyne carbon..." As written, the statement "adjacent to" is confusing. Where heteronuclear correlations are important in verifying structure, the 2D spectra should be added to the SI. As pointed out above, it would be helpful if all the NMR data were included in SI.**

- Phrase was amended to *"assigned to the alcohol methyne carbon"*
- gHSQC spectra for all compounds not previously characterized have been added to the SI.

9. **p. 12, l. 17: The rationale for assigning the quaternary carbon would be better explained by stating that no C–H cross peak was observed for the 13C signal at 95.6 ppm in the HSQC experiment.**

- Amended to "A tetrasubstituted carbon, with no correlations in the gHSQC, at 95.6 ppm was consistent with the tertiary nitrate ester."

10. **p. 16, l. 12: "Epimer" is misused here. Platydiol (the name "platydiol" automatically designates the the di-endo geometry of the diol) has a plane of symmetry, and there are no optical isomers.**

- Replaced "its epimer" with "its *trans* diastereomer." We note that the *trans*-diastereomer is epimeric at one stereocenter from platydiol.

11. **p.17, l. 14: In discussing the structural verification of nitrate ester 27, it would be helpful to include the HSQC spectrum. As presented in the text, there is not enough data to be convincing regarding the structural assignment.**

- gHSQC spectra for all compounds not previously characterized have been added to the SI.

12. **12. p. 17, bottom: The table of spectral data should have the heading at the top.**

- table heading moved to the top.

---

## Author Comment (AC4) · 12 Nov 2019

**Response to Referee #4**
Thank you for the valuable comments and suggestions. Our answers are in bullet points below the original referee comments in **bold**, changes to the manuscript are in *italics*.

**The manuscript presents synthesis methods for nitrate esters with relevance to terpene oxidation products in the atmosphere. The authors report preparation and purification of nine nitrate esters and additionally provide some details on unsuccessful methods. The information contained in this manuscript will be of interest to atmospheric chemists looking into nitrate radical oxidation products and I recommend acceptance after the following comments are addressed.**

Thank you for the general comments on the importance of our work.

1. **Was the HPLC listed on page 3 line 12 used on the Orbitrap? If so, what conditions were used (solvents, gradient, spray conditions, etc.). Were both positive and negative ion mode used? How was the instrument mass calibrated?**
   - The HRMS data was generated by direct infusion, bypassing the HPLC. The ion mode is indicated as either "ESI+" or "ESI-" as well as in the charge of the masses observed. The facility technician loaded the most recent tune files (for both positive and negative modes) and checked the calibration with a standard sample (reserpine).
2. **Acronyms I noticed that should be added to the list: CV, TLC, br, dq and qd, vs, Rf etc.**
   - abbreviations have been added to section 2.1.
3. **Exact mass measurements were only provided for a subset of the samples. Why is this the case? Also, the expected exact masses for the anions are incorrect, check to make sure the mass of the electron has been added. The expected exact mass for C10H16O4N- is 214.10848. (page 9 line 6 and page 10 line 16)**
   - HRMS data has been added for all previously unreported compounds.
   - exact mass for C10H16O4N- was corrected to 214.10848.
4. **Methanolysis products were observed when methanol was used as the analysis solvent (page 18). Were these the only products observed or were the desired products also observed? Did this occur for all the nitrate esters? Also, please clarify what M+ is for each of the masses. I can see how m/z 185.1 is formed but I do not see where m/z 223.13 is coming from. Also, are these exact mass measurements? If so, providing more numbers after the decimal is a good idea.**
   - These were low resolution GC-MS data. The *m/z* 223.1 is consistent with M+K (it was incorrectly indicated as M+Na).
5. **Figure 7 is very crowded, especially at the far right side on top. I recommend providing a little more space between compounds 22, 23 and 24 so that they are easier to see.**
   - added spacing to figure 7 as allowed by document margins.
6. **Incomplete citation on page 2, line 10**
   - replaced with "Nozière et al., 2015"
7. **m/z should be italic throughout**
   - corrected to italicized *m/z*.
8. **Formatting on the citations looks a little odd, especially the spacing between the comma and the dates**
   - Deleted extra spacing in citations.

---

## Author Comment (AC5) · 12 Nov 2019

**Referee #5**

Thank you for the valuable comments and suggestions. Our answers are in bullet points below the original referee comments in **bold**, changes to the manuscript are in *italics*.

The paper by McKnight et al. presents successful synthetic routes to the preparation of the hydroxy nitrates that are produced in nature by the atmospheric oxidation of monoterpenes. In recent years it has been shown that formation of these organic nitrates are an important sink for NOx in the atmosphere, as well as an important mechanism for the production of atmospheric aerosol. Such aerosol results from an interaction between anthropogenic NOx, and biogenic monoterpenes. Thus, interestingly, control of NOx emissions (from combustion) can influence the role of production of atmospheric aerosol from biogenic emissions. There is thus considerable interest in these compounds, and their atmospheric chemistry, and, e.g. the ability to study their hydrolysis rates. However, the atmospheric chemistry community has not had much success in making these compounds, and that is a major stumbling block to progress. Here is a paper that connects the organic synthesis community to the atmospheric chemistry community in a useful way, and the atmospheric chemistry community increasingly needs this connection, in more general terms. This paper is technically important, and will stimulate a great deal of badly needed research, likely to start with atmospheric chemists at Reed. So, this paper should be published, and I think can be done with only editorial changes. That said, the authors need to be aware that the audience here is indeed atmospheric chemists, who are not versed in the editorial jargon of synthetic organic chemists. So, the paper could be made more readable and useful if the authors shed the notion that they are writing to organic chemists. They are, at best, writing to organic chemists existing with the labs of atmospheric chemistry professors.

Thank you for the assessment of the need for this work. In addition to filling the need for nitrate ester standards, we hope that our work will help bridge the gap between the synthetic and atmospheric communities. We attempted to mitigate the use of jargon and tried to write to a more general audience. We appreciate your input and comments for improving the manuscript.

- 1. The last sentence of the Introduction is unclear 30-40% of MT emissions are at night, or the nitrate radical consumes 30-40% of the emitted MTs at night? Please note that the OH radical oxidation pathway is still an important source of MT-nitrates, e.g. as discussed in Pratt et al., 2012.
  - Clarified page 1, line 20: Replaced "This nitrate oxidation pathway has been shown to be important with 30-40% of monoterpene emissions occurring at night (Pye et al., 2010)."
  - "Nitrate oxidation pathways have been shown to be important particularly during nighttime. A large portion (30-40%) of monoterpene emissions occur at night (Pye et al., 2010). These emissions can then react with NO3 radicals, formed from the oxidation of NO2 emissions by O3, (Pye et al., 2010). "
- 2. Line 21, page 2 is there a better reference for this quite general organic chemistry laboratory hazard?
  - Despite the well-known hazards of nitric acids, explosions and other accidents occur nearly every year. I believe that it is important to reference examples of recent accidents, such as the incident at UC Berkeley.
  - A second reference (Parker, 1995) was also added.
- 3. The Ma et al. paper should not be cited. If you go to the ACP web site, it says "This paper has been retracted."
  - The reference to the original Ma paper was removed. The citation for the retraction remains.
- 4. Page 3, line 17 what is CPBA? (Good example of my comment above; most organic chemists will know that it doesn't stand for Certified Professional Behavioral Analyst).
  - We added meta-chloroperoxybenzoic acid (m-CPBA) to the list of abbreviations.

- 5. Page 3, line 27 should the reference be Crocker and Grayson, 1969? And on Page 4, line 8.
  - Reference for trans-carene oxide (page 3, line 27) is correct.
  - Page 4 line 8 was corrected from Croker, 1969 to Crocker and Grayson, 1969.
- 6. Page 5, line 8 should be Steiner et al., 2002. And line 16.
  - replaced "Steiner" with "Steiner et al., 2002"
- 7. Page 7, at 2.3.1 title, refer the Table 1 for structures. The reader needs to see the structures. Line 2 do you mean the trans-3-carene epoxide? Also on line 10, the epoxide?
  - Inserted structures for each experimental procedure.
  - Corrected "trans-3-carene" to "trans-3-carene oxide"
- 8. Page 8, line 4, those percentages are the yields? Please try to avoid organic synthetic chemists shorthand. Also line 13. For structures 16 and 17, line 22, are there yields?
  - Replaced "(53% in dioxane; 62% in DCM)" with "(53% yield in dioxane; 62% yield in DCM)"
  - The yields for 16 & 17 were reported in their individual descriptions. Added %yields in the summary description.
- 9. Page 8, line 24, what is "the title compound"? Also page 9, lines 4, 17, and 23. There is sloppy shorthand on this page, e.g. line 13 "the following". What following?
  - This nomenclature and formatting is standard for organic chemistry experimental procedures.
  - Deleted "the following" and replaced the title compound with "nitrate ester," where indicated.
- 10. Line 11 you mean the a-pinene oxide? Line 17 by "title compound" you mean the aldehyde? Similarly, I find line 23 confusing. You synthesized the alcohol?
  - Page 9, line 11, replaced with a-pinene oxide.
  - Clarified by replacing the title compound with the appropriate nitrate ester.
- 11. Page 10, lines 13 and 14 "The title compound", when the title compound is "Nitrate ester 23" does not make for great/easy reading. Perhaps it would be helpful that in each case, the structure should be shown next to the name, as a graphic? That would really make the paper more readable.
  - Clarified by replacing the title compound with the appropriate nitrate ester and added a graphic with the structure of the compound
- 12. Page 14, line 8 provide a reference for the fast hydrolysis rates.
  - Added citations to Boyd et al., 2015; Darer et al., 2011; Liu et al., 2012.
- 13. Page 19, line 6, re decomposition of ester 14 does this depend on water content in the solvent? Do you know what the decomposition products are? The corresponding diol?
  - Nitrate ester 14 was stored as a neat (i.e. water and solvent-free) liquid.
  - Decomposition produces a complex mixture of products that have not been identified.
  - Rephrased to clarify "For example, tertiary nitrate ester 4 is particularly stable stored as a solid at 0 °C. In contrast, when secondary nitrate ester 14 was stored as a neat oil at 0 °C, it decomposed to a complex mixture within a few weeks."
- 14. Page 19, line 9 the "the" methods, but "successful methods".
- corrected to "successful methods"
- 15. Line 17 insert "for" after "important".
- corrected to "is important for"